# The global implications of a Russian gas pivot to Asia

Steve Pye [1] ✉, Michael Bradshaw [2], James Price[1], Dan Zhang [1], Caroline Kuzemko[3], Jack Sharples [4], Dan Welsby [5] & Paul E. Dodds [1,5]

Recent years have seen unprecedented shifts in global natural gas trade, precipitated in large part by Russia's war on Ukraine. How this regional conflict impacts the future of natural gas markets is subject to three interconnected factors: (i) Russia's strategy to regain markets for its gas exports; (ii) Europe's push towards increased liquified natural gas (LNG) and the pace of its low carbon transition; and (iii) China's gas demand and how it balances its climate and energy security objectives. A scenario modelling approach is applied to explore the potential implications of this geopolitical crisis. We find that Russia struggles to regain pre-crisis gas export levels, with the degrees of its success contingent on China's strategy. Compared to 2020, Russia's gas exports are down by 31–47% in 2040 where new markets are limited and by 13–38% under a pivot to Asia strategy. We demonstrate how integrating energy geopolitics and modelling enhances our understanding of energy futures.

Since 2021, consumers and politicians have received a crash course in gas market dynamics and geopolitics. Following the COVID-19 pandemic, gas prices in Europe rose throughout the second half of 2021 (Fig. 1) as Gazprom, the state-owned Russian gas company, declined to refill its storage facilities or offer greater volumes to the spot market and the global demand for liquefied natural gas (LNG) exceeded supply. Russia's invasion of Ukraine in February 2022 then precipitated a dramatic reduction in Russian pipeline gas supplies to Europe to around 20% of their pre-war level (Fig. 1)[1]. This resulted from Gazprom withdrawing from the European spot market and suspending supplies to holders of long-term contracts that refused to switch to payments in roubles, while the Yamal-Europe pipeline from Russia to Germany was closed due to sanctions and the Nord Stream 1 and 2 pipelines ceased to operate after explosions.

In May 2022, the European Union announced the REPowerEU programme with the goal of independence from Russian fossil fuels by 2027. Short-term measures to reduce gas demand and manage security of supply were implemented[2–4], including rapidly increasing European LNG import capacity[5,6]. Gas prices reached record levels in the summer of 2022, but then effective policy implementation, mild winter weather, reduced Asian gas demand and a well-functioning global LNG market avoided very high prices in Europe in winter 2022–23. Nevertheless, EU Governments still earmarked over €540bn protecting consumers from the record high gas prices[7]. Europe also priced other LNG-importing countries, such as Pakistan and Bangladesh, out of the market, causing local energy shortages and some switching back to coal. Continued price volatility is likely until 2026[8], when a surge in new LNG supply, principally from the US and Qatar, could lower global prices, potentially enabling Europe to secure imports at lower prices as it shuns Russian gas[9,10], including that supplied by LNG[11–14].

We explore three implications of these events. First, Russia is losing most of its largest and most lucrative pipeline gas export market. Europe accounted for 66% of pipeline exports in 2021 so Russian government revenue, initially protected by higher prices, has reduced substantially[15,16]. To compensate, Gazprom aims to expand its exports to Asia[17]. Second, Europe is increasingly dependent on LNG imports so is exposed to the liquidity and volatility of the global LNG market. EU policies to accelerate gas demand reduction could reduce this dependence. Third, whether Russia can pivot to Asia has wider consequences for global gas security as it will influence the scale of China's call on the global LNG market (reducing competition for Europe)[18].

[1]UCL Energy Institute, University College London, London, United Kingdom. [2]Warwick Business School, University of Warwick, Coventry, United Kingdom. [3]Department of Politics and International Studies, University of Warwick, Coventry, United Kingdom. [4]Oxford Institute for Energy Studies, Oxford, United Kingdom. [5]UCL Institute for Sustainable Resources, University College London, London, United Kingdom. ✉ e-mail: s.pye@ucl.ac.uk

This research is a collaboration between social scientists interested in energy geopolitics and global gas security and energy systems modellers focused on quantifying the role of fossil fuels in the energy transition[19]. Our approach employs a geopolitical framing to inform the development of scenarios about Russia's future role in global gas markets, the implications of which are assessed using the TIAM-UCL Integrated Assessment Model (IAM) (see "Methods"). Among a number of aspects, it is this interdisciplinary approach, a dynamic process of creating and modelling scenarios that endeavour to capture key features of today's energy geopolitics, that brings a distinctive approach to this study. This allows for both a qualitative and quantitative exploration of the interrelationships under different Russian gas supply futures shaped by recent geopolitical events.

We find that even under a pivot to Asia strategy, Russia struggles to regain pre-crisis gas export levels in all scenarios. Compared to 2020, Russia's gas exports are down by 13–38% in 2040, and by 31–47% where new markets to Asia are limited. A higher demand from China

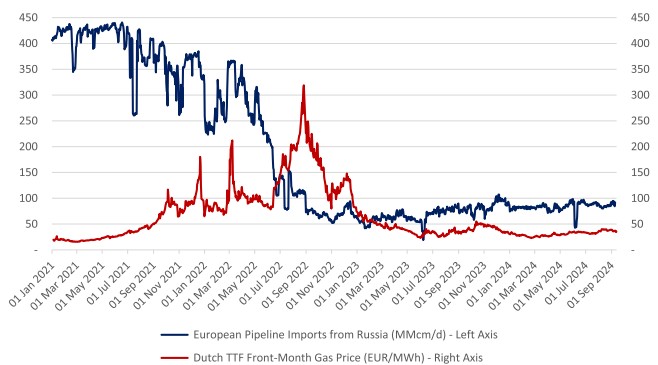

**Fig. 1 | European pipeline imports from Russia (mmcm/d) and Dutch TTF Front-Month Gas Price (EUR/MWh).** Data from ENTSOG Transparency Platform[49] and Argus Direct[50]. This definition of Europe excludes Turkey.

does not significantly improve prospects for Russia. Crucially, any future pivot to Asia is contingent on Chinese energy security and climate mitigation strategy and objectives. Alongside Europe's supply-side strategy for future gas demand, such decisions are likely to be consequential for the global LNG market, and future price volatility.

Pipeline gas has been exported from Russia to Europe for almost 60 years[20]. Exports started in the 1960s and expanded through the 1970s and 1980s as Western Europe invested in gas-for-pipe deals to bring gas from West Siberia[21]. New transit states emerged in 1991 as the Soviet Union fragmented. As disputes between Russia and Ukraine in the 2000s resulted in gas shortages, and as Gazprom abused its monopoly position in central Europe and the Baltic states, Europe's dependence on Russian gas became a strategic concern. The EU's response was to align Russia's gas trade with its wider competition policy and to create a single integrated energy market[22,23]. Russia's response was to invest in new offshore pipelines to by-pass transit states (Fig. 2). The Nord Stream pipelines transit through the Baltic Sea to land in Germany[24], while TurkStream transits through the Black Sea linking directly to Turkey. In 2021, 40% of gas consumed in the EU was sourced from Russia via pipeline.

Russia also sought to develop new fields in the Far East to access China as a new export market[25] through its Eastern Programme in 2007[26] and through government action against Western oil companies to take over the Kovytka project and gain control of the Sakhalin-2 project[27]. In 2014, it agreed a $400 billion deal to build the Power of Siberia 1 (PoS 1) pipeline from Yakutia to China (Fig. 2). It commenced exports in 2019 and should reach its 38 bcm capacity by 2025[28]. A second 50 bcm, 3,550 km pipeline, Power of Siberia 2 (PoS 2), has been proposed to deliver gas from the Yamal Peninsula, which currently supplies Europe[29]. Agreement is yet to be reached. In 2022, a deal was signed for a new 10 bcm pipeline, to be completed by 2027, to take Sakhalin gas to China by extending the pipeline that supplies Vladivostok[30].

Export prices have not been published but it is highly unlikely that trade with China is as profitable as the trade that Russia has lost with

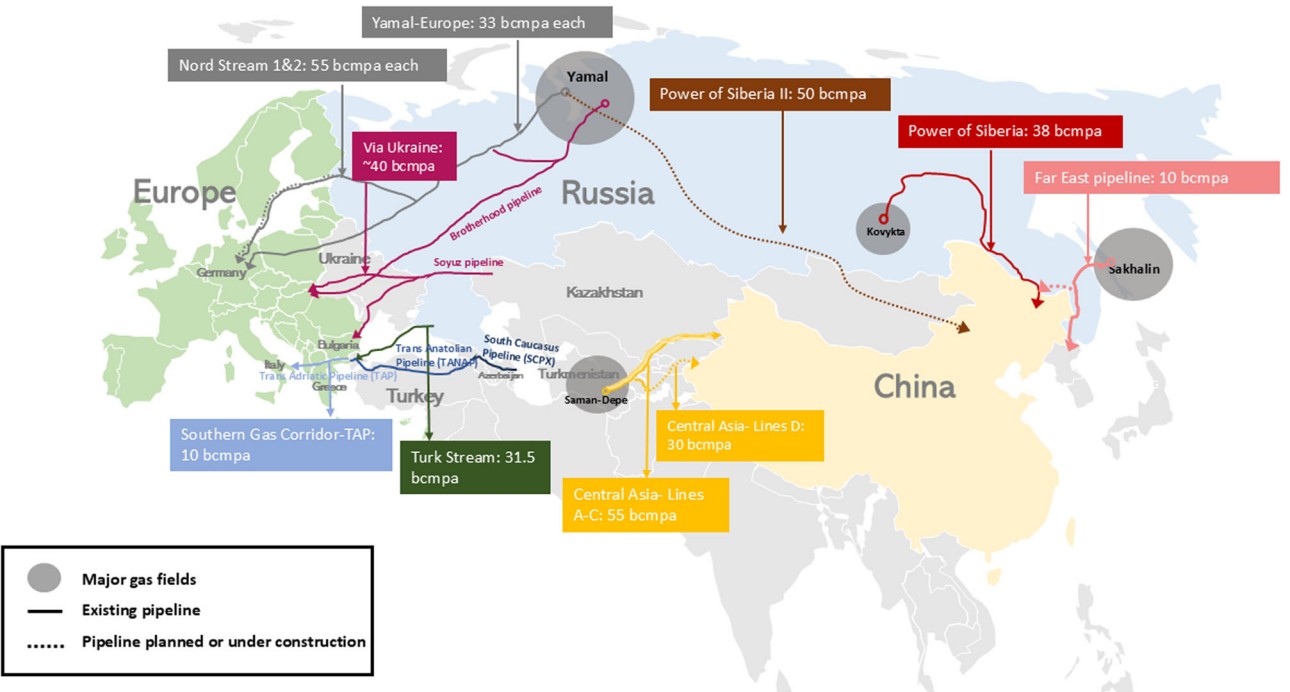

**Fig. 2 | Eurasian natural gas pipeline network.** This map shows the main pipeline routes and current capacities in the Eurasian region, covering Europe, Russia, China and Central Asia. The map was constructed using pipeline information exclusively from GEM Wiki[51], with the base map sourced from freeworldmaps.net, at https://www.freeworldmaps.net/powerpoint/.

**Table 1 | China's current and projected gas import capacity via pipelines**

| Export country / region | Cross-border pipelines | Start year | Capacity (bcm/year) | Status |
|---|---|---|---|---|
| Myanmar | China-Myanmar Pipeline | 2013 | 12 | Operational but with actual sales of 3–4 bcm |
| Central Asia | Central Asia-China Pipeline A | 2009 | 15 | Operational |
| | Central Asia-China Pipeline B | 2010 | 15 | Operational |
| | Central Asia-Pipeline C | 2014 | 25 | Operational |
| | Central Asia-China Pipeline D | 2026 | 30 | Under construction |
| Russia | Power of Siberia (PoS) 1 | 2019 | 38 | Operational |
| | Far East Pipeline | 2027 | 10 | Under construction |
| | Power of Siberia (PoS) 2 | 2030 | 50 | Planned |

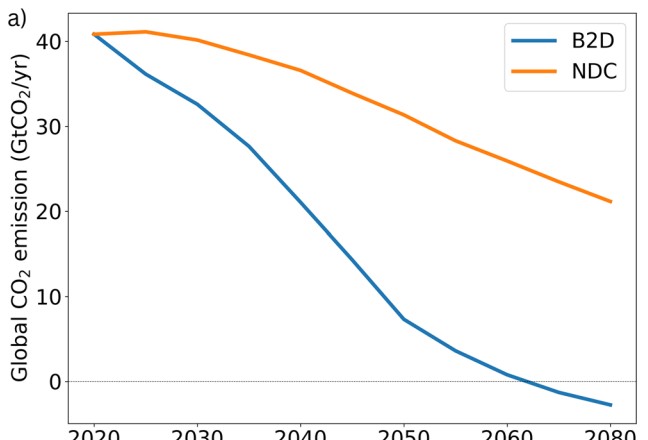
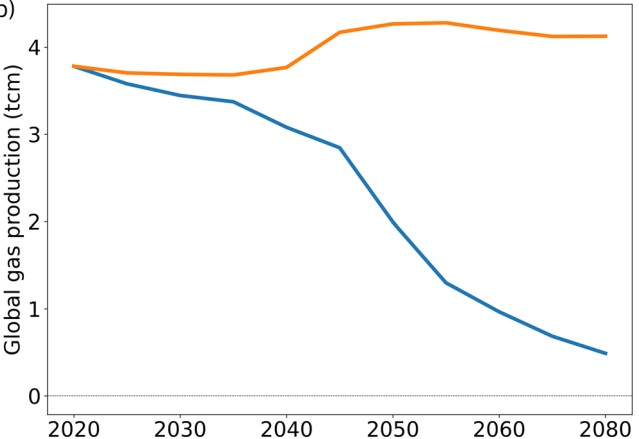

**Fig. 3 | Global CO₂ emissions and gas production under NDC and B2D scenarios.** **a** CO₂ emissions under NDC and B2D climate policy ambition levels and (**b**) corresponding modelled gas production trajectories. We note that at the global level, emissions and gas production pathways are highly comparable between LM and P2A and are therefore not differentiated. For gas export levels by pipeline and LNG, see SI Fig. 3.

Europe[15]. Russia's need to find new markets weakens its bargaining position with China[31], whilst China may be cautious of becoming too dependent on Russian pipeline gas. What happens next will have significant implications for global gas security[13].

Our analysis explores possible implications of Russia's future gas exports via two divergent but plausible geopolitical scenarios (which are further elaborated in the "Methods" section). The first, Limited Markets (LM), sees a protracted conflict in Ukraine rule out prospects for a return of flows through pipelines from Russia to Europe, resulting in a permanent halt of all gas trade (pipeline and LNG) with Russia by 2027 (as per the EU target). Alternative routes for Russian exports are limited by sanctions on LNG technology and infrastructure, slowing the ability of Russia to expand its LNG capacity in the near term[32], while EU and most other OECD countries stop taking Russian LNG cargoes. At the same time, China and Russia fail to agree to additional pipeline capacity beyond the current PoS1 and Far East pipelines (Table 1). This reflects China's position on energy security; avoiding increased dependency given the Russia-European supply crisis, and a focus on further diversification based on domestic production and LNG.

The second, Pivot to Asia (P2A), sees a cessation of the Russia-Ukraine war and agreement between the two sides in the mid-2020s, albeit with a lack of resolution around Russia's occupation of territory in Eastern Ukraine and Crimea. Given this context, Europe remains unwilling to revert to its previous dependence on Russian energy, with no pipeline gas via Ukraine after 2025 due to the non-renewal of the transit agreement, and a maximum 15 bcm via TurkStream. Russia, therefore, pivots towards China, almost doubling pipeline capacity by 2040 through the construction of PoS 2. With fewer restrictions on expansion and supply, Russia expands LNG exports, including to Europe. A third counterfactual case (REF) highlights how a world

without the Russian invasion of Ukraine might have played out. This is insightful to highlight the importance of geopolitical factors.

While LM and P2A explore how the flow of gas is determined by the interplay between regional and global demand, this demand is subject to high levels of uncertainty in the medium-to-long term. A key factor influencing global gas demand is the pace of the energy transition, which is driven by the stringency of climate policy. To simulate this, two levels of policy ambition are introduced for each scenario; a current policy, or National Determined Contributions (NDC) case, where countries meet policy ambition reflected in their current NDCs, and a below 2 degrees (B2D) case, where average global temperature rise is kept below 2 °C (see "Methods"). The resulting global gas demand levels can be seen in Fig. 3. Combining the three scenarios (including counterfactuals) with the two climate ambition levels results in six main scenarios (which are also supplemented by sensitivity cases, see SI section 3).

## Results

### A challenging outlook for Russian exports
In both geopolitical scenarios (LM and P2A), Russian gas exports in 2030 are substantially lower than pre-war levels (Fig. 4a, b). Recovery to 86–96% of 2020 levels is observed in the long term only in cases where global demand is maintained, that is under the NDC cases (see Fig. 3b). In P2A, this is achieved through expansion of pipeline capacity to China and an increase in LNG exports. In LM, without pipeline expansion to China, exports are dependent on LNG exports increasing to 120 bcm by 2050. However, this expansion of LNG may be challenging to achieve if Western sanctions persist (see SI section 3.1). Compared to the counterfactual (REF_NDC), which would have seen exports maintained at 2020 levels over the next 25–30 years, Russia's

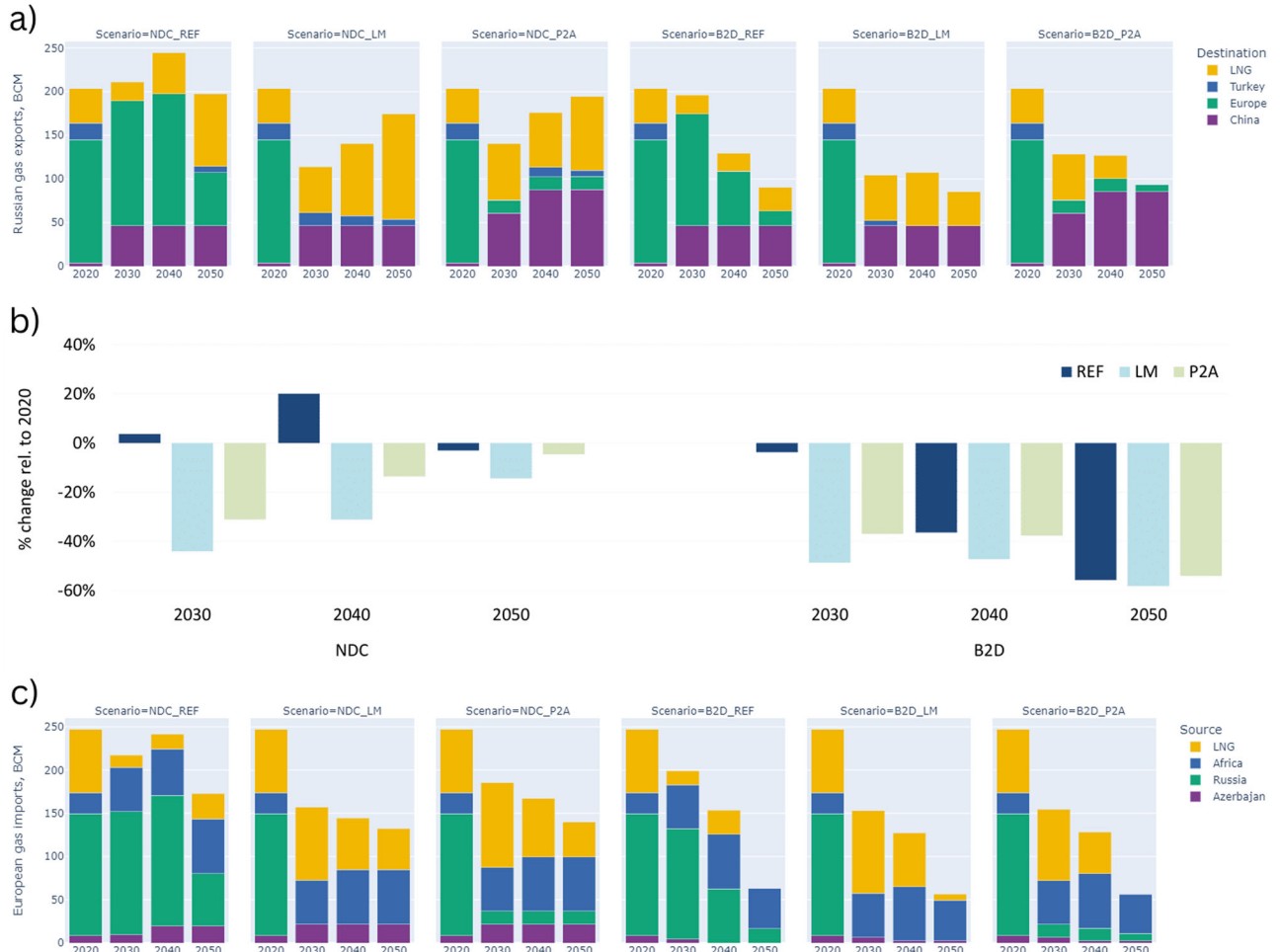

**Fig. 4 | Russian gas exports and European gas imports by scenario, 2020–2050.**
**a** provides information on Russian pipeline exports by destination plus LNG
exports. **b** shows the relative percentage change in total Russian export levels
relative to 2020. **c** provides information on European pipeline imports by region of
origin plus imports via LNG routes. The first part of the scenario label denotes the

climate policy ambition, *NDC* or *B2D*. The second part of the label denotes the
geopolitical scenario, *LM* or *P2A*. The counterfactual for both *LM* and *P2A* only has
the climate policy ambition level in its label. Note that *European* refers to con-
tinental Europe, so excludes the UK. Data for (**a**) and (**b**) can be found in SI Table 4,
and for (**c**) in SI Table 5, and in the file supplementary data 1.

gas exports are down by 13–31% in 2040 under P2A and LM respec-
tively, compared to 2020, and cumulatively reduced by 21–32% over
the period to 2050, compared to the counterfactual (SI Table 4).

In contrast to the NDC scenarios, shrinking global demand in the
B2D scenarios causes a steep, long-term decline in Russian exports
(Fig. 4b). The war with Ukraine expedites this decline, as seen by lower
export levels under LM and P2A in 2030, relative to the counterfactual,
REF_B2D. The pivot to Chinese markets via increases in pipeline
capacity sustains export levels in the P2A case post-2030, similar to
those observed in REF_B2D; however, in LM, levels are lower and only
buoyed by an increase in Russian LNG exports.

Crucially, a sensitivity case with projected higher demand in China
does not alleviate this export decline in the LM and P2A B2D cases, nor
in the NDC cases. The analysis highlights that this additional Chinese
demand is met by other LNG suppliers, not Russia, and supplemented
by a limited increase in pipeline gas from Central Asia (see SI
section 3.1).

**Europe's gas demand in terminal decline**
With Russian gas imports to Europe greatly reduced, the region con-
tinues to reduce and diversify its supply. Differences between LM and
P2A are less pronounced for this region due to similar constraints on
Russian gas pipeline imports, except for some flows through TurkStream
in P2A. With a drop in imports through the 2020 s, Europe sees an

increased reliance on LNG imports and greater utilisation of pipelines
from Africa. The overall decline in total imports is gradual under the NDC
case (2.1% per annum) but much more rapid in B2D (4.8%), reflecting the
impact of climate policy. It is of note that B2D_REF also sees a strong
decline out to 2050 albeit at a slightly slower rate of 4.5% per annum.
This suggests that under robust climate policy, Russian exports were at
risk of contraction in the long term even without the recent crisis, albeit
not at the rapid rate of decline expedited by the crisis.

With limited prospects for increasing domestic production in
future years, an increased dependency on LNG in LM and P2A is
observed (Fig. 5). This means an increased exposure to global supply
pressures from increased market demand, for example from China. A
sensitivity case, where China increases its own LNG imports due to
lower domestic production, in a world of higher gas demand (NDC)
shows Europe consequently reducing its imports of LNG due to price
pressures (SI Figure 9). This suggests that stronger diversification
alongside reduction in gas demand could help reduce exposure to
price increases driven by growth in gas demand in other emerging
markets.

**Chinese import strategy is key to the future Russian
export market**
Prospects for Russian exports are not only impacted by limits to Eur-
opean markets but also by future trading arrangements with China,

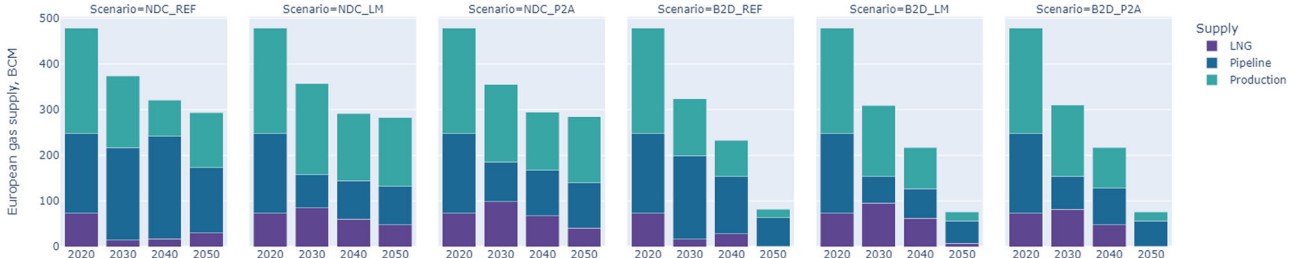

**Fig. 5 | European gas supply, 2020–2050.** This figure shows the origin of gas supply to Europe, with 'production' produced domestically, and imports via 'pipeline' and 'LNG'. The first part of the scenario label denotes the climate policy ambition, *NDC* or *B2D*. The second part of the label denotes the geopolitical scenario, *LM* or *P2A*. The counterfactual for both *LM* and *P2A* only has the climate policy ambition level in its label. Note that *European* refers to continental Europe, so excludes the UK. Data for this figure can be found in the file supplementary data 1.

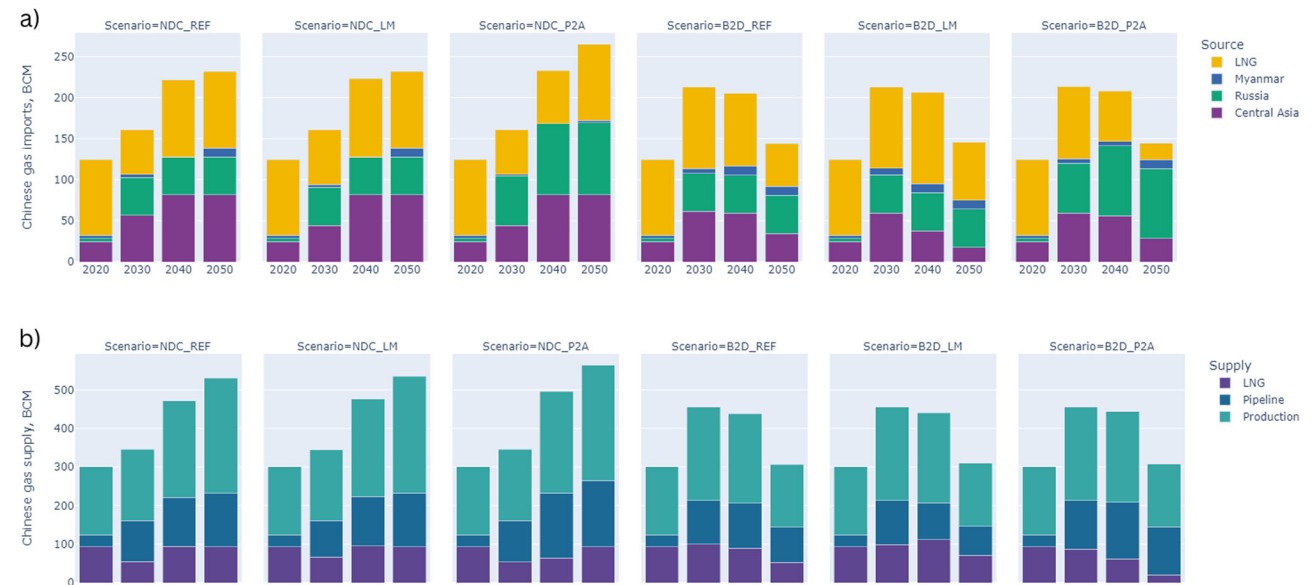

**Fig. 6 | Chinese gas imports and total gas supply, 2020–2050. a** provides information on Chinese imports by origin plus total LNG imports. **b** shows the origin of gas supply to China, with 'production' produced domestically, and imports via 'pipeline' and 'LNG'. The first part of the scenario label denotes the climate policy ambition, *NDC* or *B2D*. The second part of the label denotes the geopolitical scenario, *LM* or *P2A*. The counterfactual for both *LM* and *P2A* only has the climate policy ambition level in its label. Further data on China's gas balance can be found in SI Table 6, and in the file supplementary data 1.

through existing and new pipeline capacity (see Table 1). Such prospects will be determined both by the rate of growth in China's gas demand, and how it chooses to meet that demand. Its future supply strategy will likely be informed by a combination of energy security, economic and climate objectives; Russian pipeline supplies may be cost-competitive and protect China from the risk of LNG price volatility but would inevitably lock China into greater dependency on Russian supply. Levels of domestic shale gas production would also have a significant bearing on required import levels.

In the high global demand case (NDC), China's gas demand grows from 300 bcm in 2020 to 530–563 bcm by 2050 (Fig. 6b, SI Table 6). This is driven by growing demand in the industrial sector, with some limited increase for power generation given that most growth in electricity demand is met by renewable sources. Gas import increases are via pipeline and LNG, with Russian pipeline gas offering cost advantage over other options, with full utilisation when available.

In B2D, gas imports see stronger growth in the near term, due to the push to remove coal from the industrial sector under carbon constraints (Fig. 6b). This means a stronger reliance on LNG, and existing pipeline capacity from central Asia. Once built and available, Russian pipeline supply is the preferred import route as demand declines post-2030. A key assumption underpinning these scenarios is

that domestic gas production grows in line with Chinese policy, maintaining its share of total supply at over 50% out to 2050. Sensitivity cases highlight that much lower domestic production would lead to an increased reliance on LNG, and lead to an overall reduction in total gas supply (SI sections 3.4, 3.5). This could have implications for wider gas markets, as indicated for Europe.

## Discussion

Our analysis highlights the globally connected nature of gas supply and demand, and how geopolitics can shape the energy transition. The impacts of the European gas crisis are most stark for Russia and Gazprom. Without the option to build new pipeline capacity to China, expanding LNG becomes a key strategy – but even in a high demand world, Russia cannot recover pre-crisis export levels. Export levels only recover substantively if PoS2 is built, and LNG capacity expands. However, three factors could undermine such prospects. First, China has a strong negotiating position to agree to a lower price than was available to Russia when exporting to Europe, meaning that even if gas exports were increased, revenues would not recover. This stronger position is based on China's diversified pipeline supply and large LNG import potential (based on physical capacity and contracting). Second, global reduction in gas demand driven by climate action compounds

**Table 2 | European gas pipeline import constraints in geopolitical scenarios**

| From | Pipeline | Scenario | 2020 | 2025 | 2030 | 2035 | 2040 |
|---|---|---|---|---|---|---|---|
| Russia | Nord Stream I[a] | Base | 55 | 55 | 55 | 55 | 55 |
| | | LM | 55 | 0 | 0 | 0 | 0 |
| | | P2A | 55 | 0 | 0 | 0 | 0 |
| | Yamal | Base | 33 | 33 | 33 | 33 | 33 |
| | | LM | 33 | 0 | 0 | 0 | 0 |
| | | P2A | 33 | 0 | 0 | 0 | 0 |
| | Via Ukraine[b] | Base | 40 | 40 | 40 | 40 | 40 |
| | | LM | 40 | 15 | 0 | 0 | 0 |
| | | P2A | 40 | 15 | 0 | 0 | 0 |
| | TurkStream | Base | 31.5 | 31.5 | 31.5 | 31.5 | 31.5 |
| | | LM | 31.5 | 10 | 0 | 0 | 0 |
| | | P2A | 31.5 | 15 | 15 | 15 | 15 |
| Africa | Greenstream | All | 11 | 11 | 11 | 11 | 11 |
| | Trans-Mediterranean | | 33.5 | 33.5 | 33.5 | 33.5 | 33.5 |
| | Medgaz | | 8 | 8 | 8 | 8 | 8 |
| | Maghreb–Europe | | 12 | 12 | 12 | 12 | 12 |
| Azerbaijan | Southern Gas Corridor[c] | All | 10 | 20 | 20 | 20 | 20 |

[a]Following the explosions that disrupted its operation in 2022, Nord Stream II never comes online
[b]Brotherhood and Soyuz pipelines
[c]Southern Gas Corridor transports gas from Azerbaijan to Europe, connecting with South Caucasus Pipeline (SCPX), Trans Anatolian Pipeline (TANAP), Trans Adriatic Pipeline (TAP).

the challenge of recovering export levels. Even without the gas crisis, our counterfactual scenario shows that Russian exports would decline due to climate policy, albeit at a slower rate. Thirdly, sanctions could thwart the ability of Russia to expand its LNG export capacity.

Europe has responded to the crisis by both increasing LNG imports and reducing overall demand for gas. LNG imports are set to continue growing out to 2030, with no or limited Russian gas pipeline imports. This growth in LNG demand, alongside that from China where pipeline capacity expansion from Russia is limited and domestic production is reduced, could increase prices. In an alternative scenario with better prospects for Russian gas pipeline exports to China (P2A), Chinese demand for LNG is lower. This could ease cost pressures on Europe due to a more liquid LNG market. Post-2030, gas demand in Europe is set to decline as a result of the REpowerEU policy (NDC) and net zero targets (B2D), which would reduce LNG demand and create risks of over-capacity and stranding of LNG infrastructure in Europe. This trend suggests that Russian exports to Europe were vulnerable even in the absence of crisis. Our modelling suggests that geopolitical events and policy responses have reinforced and sped up that trend.

A key element of Russia's strategy is expanding pipeline capacity to China. However, this is contingent on Chinese strategy, as it balances its climate policy objectives and energy security concerns, and the outlook for its domestic production. With a stronger negotiating hand, PoS2 would be a cheap source of gas; however, China will be cognisant of Europe's experience and may prefer a more diversified supply through LNG, accepting price volatility risks. If China were to take a more diversified supply option, this will have implications for the LNG market, with potential for market tightening, although of course this will also be dependent on large suppliers such as USA and Qatar. There is uncertainty too around Chinese imports. There is no guarantee that higher Chinese import demand (driven by higher economic growth or lower domestic production) would lead to an increased market for Russia, which we find would be met by other LNG suppliers. Any contraction in Chinese imports will make for an even more challenging outlook for Russia.

The use of geopolitical frames to drive model-based scenarios illustrates the complex interrelationships between climate action and energy security in the face of the geopolitical upheaval caused by

Russia's invasion of Ukraine. In all scenarios, Russia has reduced exports and the Russian Government has reduced revenues. The differences between scenarios are in the scale of loss of that export market, both in value and volume terms. Building PoS2 compensates for some loss of markets in Europe, but Russia becomes more dependent on China and loses geopolitical influence. The global implications of the different scenarios are transmitted via the global LNG market, where greater competition between Europe and China may potentially mean sustained higher prices and ongoing affordability concerns for some consumers. Conversely, greater climate action and the construction of PoS2 means less LNG demand all around and greater energy security in Europe.

Developing systematic modelling is a first step in beginning to explore and map out potential futures for the global gas market, whilst most geopolitical analyses fail to do so. In so doing, we enhance our understanding of the complex relationships between different actors on both sides of gas demand and supply equation. We also demonstrate the benefits of working to operationalise energy systems modelling and bring it together with the study of energy geopolitics to provide a more interdisciplinary view of potential energy futures. We argue that such efforts to bridge the integration gap we identified previously are essential to better understand and support the energy transition[19].

## Methods
### Approach to scenario definition
Scenarios have been developed for this research based on interaction between economic geography and political science disciplines, with energy systems modellers. This is motivated by a recognition that such communities need to collaborate to better reflect energy geopolitics in scenarios. By energy geopolitics, we mean 'the interaction of geographical factors, such as the distribution of centres of supply and demand, with state and non-state actors' attempts to ensure an affordable, reliable and sustainable supply of energy'[19].

We consider two plausible scenarios for the future of natural gas geopolitics between Russia, Europe and China and how this shapes gas trade and supply. Our Limited Markets (LM) scenario assumes that protracted conflict in Ukraine results in continued decline in gas

**Table 3 | Chinese gas pipeline import constraints in geopolitical scenarios**

| From | Pipeline | Scenario | 2020 | 2025 | 2030 | 2035 | 2040 |
|---|---|---|---|---|---|---|---|
| Russia | Power of Siberia I | Base | 10 | 38 | 38 | 38 | 38 |
| | | LM | 10 | 38 | 38 | 38 | 38 |
| | | P2A | 10 | 38 | 38 | 38 | 38 |
| | Power of Siberia II | Base | 0 | 0 | 0 | 0 | 0 |
| | | LM | 0 | 0 | 0 | 0 | 0 |
| | | P2A | 0 | 0 | 25 | 25 | 50 |
| | Far East Pipeline (Sakhalin) | Base | 0 | 0 | 10 | 10 | 10 |
| | | LM | 0 | 0 | 10 | 10 | 10 |
| | | P2A | 0 | 0 | 10 | 10 | 10 |
| Central Asia | Central Asia Line A-C | All | 55 | 55 | 55 | 55 | 55 |
| | Central Asia Line D | All | 0 | 0 | 30 | 30 | 30 |
| Myanmar | Myanmar Pipeline | All | 12 | 12 | 12 | 12 | 12 |

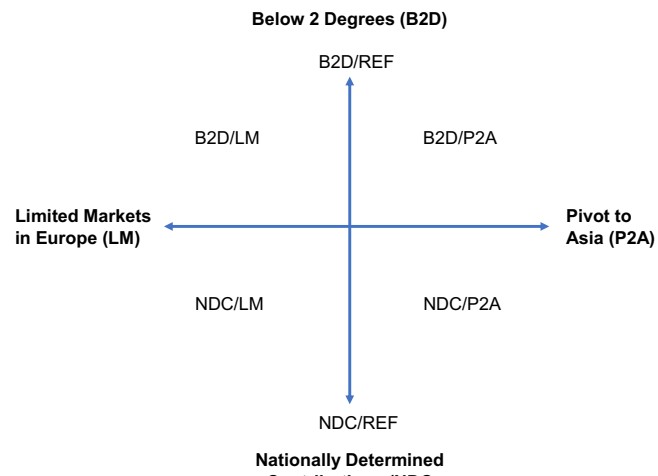

**Fig. 7 | Scenario matrix of geopolitical and climate ambition dimensions.** The horizontal axis shows the geopolitical dimensions while the vertical axis shows the level of climate ambition, used to derive a spread of gas demands. Four core scenarios are derived based on the combination of geopolitical and climate dimensions, with two additional scenario (REF) constructed to provide case without geopolitical factors recognised.

exports to the EU, and challenges for Russia in establishing alternative export routes via LNG or pipelines to Asia. Our Pivot to Asia (P2A) scenario sees Russian pipeline exports to Europe also declining from 2022 levels and failing to recover to pre-conflict levels. This results in Russia pivoting towards China, with the shortfall in lost European demand partially offset by constructing the Power of Siberia 2 pipeline. All scenario assumptions on pipeline capacities are provided in Table 2 and Table 3.

The two geopolitical scenario narratives are described below. We then describe two international climate policy regimes under which these two scenarios might play out. These are a regime where countries meet their NDC commitments but do not go beyond this level of ambition, and a regime which establishes policies that keep average global temperature rise well below 2 °C. The resulting four scenarios are then compared to a counterfactual case, again using these two climate policy regimes, in which we imagine a situation without the geopolitical narratives used in LM or P2A. This results in a further two base or REF scenarios, one per climate future (see Fig. 7).

Finally, we have also undertaken a sensitivity analysis to further explore some of the key assumptions in our geopolitical scenarios LM

and P2A, and determine their impact on the modelling results. These include -

- Exploring reduced Chinese domestic production, with implications for the level of gas imports China might require.
- Exploring increases in Chinese gas demand, given the uncertainty highlighted in SI section 1. We have reviewed the range of gas demand levels in China and assessed the implications of a higher level of demand in China.
- Assessing further constraints on Russian LNG exports in the medium to long term, based on the continuation of western-led sanctions.

This analysis is described in detail in SI section 3.

**Scenario narratives**

Limited Markets (LM): Under this narrative, Russian exports of gas see a sharp decline in the 2030 s due to the complete shutdown of pipeline exports to Europe by 2030, in line with the REPowerEU aim to phase out gas imports by 2027, and restrict LNG routes to Europe, Japan and South Korea. A protracted conflict ensures that the EU's resolve to improve its energy security by sourcing alternative gas supplies and reducing demand for gas is strengthened, as well as action to transition towards a low carbon energy system.

This scenario, similar to that in the World Energy Outlook (WEO) 2022 Stated Policies Scenario (SPS), sees Russian pipeline exports of around 150 bcm to Europe in 2021 drop to 25 bcm by 2025, and zero by 2030. Despite the continued absence of EU sanctions on gas, the Nord Stream pipelines (55 bcm) never come back online following the explosions that disrupted their operation in September 2022. The Yamal pipeline (33 bcm) also lies redundant, without a transit agreement in place between Russia and Poland, and with Germany resolved to seek alternative supplies. Routes through Ukraine, which supplied over 80 bcm in 2019, become increasingly challenging to operate due to the conflict and ongoing contractual disputes between Russia and Ukraine, and decline from 20 bcm in 2022 to zero flow by 2030. The TurkStream pipeline, supplying countries in Eastern Europe (notably Serbia and Hungary), maintains supply levels of around 12 bcm in 2025, but also sees supply ended by 2030.

Russian exports are limited to existing pipelines supplying the Chinese market and LNG trade with non-G7 regions. These export options do not come close to filling the gap left by the closure of the European market. Pipeline capacity to China peaks at 54 bcm by 2030, including Power of Siberia 1 which provides flows of 38 bcm by 2025 and an extension of the Far East pipeline to China. Constraints on European exports, and limited options for Russia, sees China driving down the contract prices, meaning that Gazprom revenues are also hit on a per unit basis. Under this scenario, China stalls on agreeing an

investment decision on the Power of Siberia 2 pipeline, limiting Russia's options further. This is based on Chinese energy security concerns related to increased dependence on Russian pipeline gas, given the recent European supply crisis, and a strategy towards increased diversification of supply via domestic production and multiple LNG contracts. The ability of Russia to increase its LNG capacity is constrained by western sanctions, limiting development of the fleet of LNG carriers and the technologies needed to support increased liquefaction capacity.

Pivot to Asia (P2A): Under this narrative, there is a cessation in the conflict in the mid 2020 s although disputes continue around occupied territory in the East of Ukraine and Crimea. However, this does lead to some political agreement about the continuation of Russian gas exports via TurkStream. TurkStream sees flows of up to 15 bcm into East and Southeast Europe via Bulgaria to Serbia and Hungary. There is no expansion of capacity for this pipeline route due to the bottleneck on the Serbia-Hungary border, where capacity is 8.4 bcm. Even with transit volumes via Ukraine decreasing, there is very limited spare capacity to increase Russian pipeline gas, not only compounded by infrastructure limits but countries in SE Europe diversifying away from dependency on Russian gas.

Similar to LM, transit volumes via Ukraine continue to reduce, with failure to extend the current transit agreement between Gazprom and the Ukraine TSO which is set to expire by 2025. Nord Stream and Yamal pipelines remain redundant, with Europe diversifying sufficiently, either to new sources of gas or alternative energy, meaning that restarting flows through these pipelines is not a political priority. There are no restrictions on the EU and other G7 nations taking Russian LNG cargoes.

Compared to pre-conflict levels, gas exports to Europe remain low. Russia therefore pivots towards China in search of new export opportunities. This is realised primarily through building the proposed new 2,600 km pipeline, Power of Siberia 2, taking gas from the Yamal Peninsula in western Siberia, through Mongolia to the Chinese market. This results in an additional 50 bcm per annum by 2035 (and 25 bcm by 2030). This is enabled by China seeing the new investment in pipeline capacity as a means of diversifying supply, avoiding overreliance on LNG. While Power of Siberia 2 almost offsets the loss of exports via Nord Stream in volumetric terms, the financial benefits are significantly lower due to China's ability to drive contract prices lower. China's position of strength in negotiations comes from its diversified pipeline supply, large LNG import potential and stable domestic production outlook.

### Climate policy regime assumptions

To capture the impact of different climate policy ambitions on our geopolitical scenarios, we model two potential futures. First, an NDC case has been developed from Meinshausen et al.[33] This study provides NDC derived GHG emissions targets for 2025 and 2030 for 196 countries as of November 2021, following the completion of COP26, together with pledges for emissions from international aviation and shipping at the global level for these years. The NDC targets are aggregated to the TIAM-UCL regions while the international transport emissions (aviation and shipping) are distributed to the regions using the share of these emissions by region in 2020 according to IEA data.

A key question is then how to extend out 2030 NDC targets to the model's full time horizon, i.e., what will climate policy look like beyond 2030. Here we follow van de Ven et al.[34] and assume that the average annual rate of reduction of the emissions intensity of GDP during 2020–2030 for each region continues post-2030. To do this we use GDP projections from Shared Socio-economic pathway 2 (SSP2). Finally for this climate policy case, we add on land-use, land-use change and forestry (LULUCF) emissions based on pathways that are already part of TIAM-UCL (see the model description below).

**Table 4 | Long term climate targets and pledges**

| Region | Year | Emission coverage (GHG or CO2) |
|---|---|---|
| Australia | 2050 | GHG |
| Canada | 2050 | GHG |
| China | 2060 | CO2 |
| Central and South America | 2050 | CO2 |
| Europe | 2050 | GHG |
| India | 2070 | CO2 |
| Japan | 2050 | GHG |
| South Korea | 2050 | GHG |
| UK | 2050 | GHG |
| USA | 2050 | GHG |

Second, we represent more ambitious climate action by developing a below two degrees scenario (B2D) which sees a 66% chance of limiting warming to 1.75 C based on an 800 GtCO$_2$ carbon budget from 2018[35]. In this scenario, we also include a number of net-zero pledges[34], which are applied to the regions in Table 4 in an effort to apply a dimension of equity to climate policy in B2D scenarios while also drawing on targets that have a real-world basis.

Finally, in both climate policy scenarios (i.e., NDC and B2D), we represent REPowerEU, the European Union's policy package to reduce its dependence on Russian fossil fuels and its exposure to the gas market volatility caused by the invasion of Ukraine. We implement the core target of REPowerEU, namely that the bloc should achieve a 45% share of renewable energy in final energy consumption by 2030, by drawing on European Commission modelling[36]. That effort translated this target into a 69%, 32% and 60% renewable energy share across electricity, transport and buildings, respectively, in 2030. We also include the EU NDC as of November 2021 which included a 55% reduction in GHG emissions from 1990 levels by 2030, and the energy efficiency targets associated with REPowerEU, which require a minimum of a 13% reduction in final energy consumption in the buildings and transport sectors. In addition, we also capture targets for low carbon hydrogen production and imports, biomethane production and heat production by heat pumps in buildings.

### The TIAM-UCL model

To explore the implementation of the geopolitical scenarios and their implications for natural gas supply, we use the TIMES Integrated Assessment Model at University College London (TIAM-UCL)[37,38]. This model provides a representation of the global energy system, capturing primary energy sources (oil, fossil methane gas, coal, nuclear, biomass, and renewables) from production through to their conversion (electricity production, hydrogen and biofuel production, oil refining), their transport and distribution, and their eventual use to meet energy service demands across a range of economic sectors. Using a scenario-based approach, the evolution of the system over time to meet future energy service demands can be simulated, driven by a least-cost objective. The model uses the TIMES model framework (described in SI section 3).

The model represents the countries of the world as 16 regions (SI section 3), enabling detailed characterisation of energy systems in each region and the trade flows between regions. Upstream sectors within regions that contain members of OPEC are modelled separately, so as an example, the upstream sector in the Central and South America (CSA) region will be split between OPEC (Venezuela) and non-OPEC countries. As part of this study we have also disaggregated upstream oil and gas resources in the Former Soviet Union (FSU) region to explicitly model Russian gas export scenarios (see SI section 5). Regional coal, oil and gas prices are generated within the model. These incorporate the marginal cost of production, scarcity

rents (e.g., the benefit foregone by using a resource now as opposed to in the future, assuming discount rates), rents arising from other imposed constraints (e.g., depletion rates) and transportation costs, but not fiscal regimes. This means full price formation, which includes taxes and subsidies, is not captured in TIAM-UCL, and remains a contested limitation of this type of model[39]. Further information on the model characterisation of fossil resources can be found in Welsby et al.[38] and in SI section 4.

The model has a limited number of technological options for carbon dioxide removal (CDR) from the atmosphere, including a set of bioenergy with carbon capture and storage (BECCS) technologies, in power generation, industry, and in hydrogen and biofuel production. The primary limiting factor on these technologies is the global bioenergy resource potential, which is set at a maximum 112 EJ per year, in line with estimates from the UK Committee on Climate Change (CCC) biomass report[40]. This is a lower level than the biomass resource available in many other integrated assessment scenarios for 1.5 °C (which can be up to 400 EJ/yr)[41,42], and is more representative of an upper estimate of the global resource of truly low-carbon sustainable biomass based on many ecological studies[43].

TIAM-UCL also includes $CO_2$ emissions from LULUCF at the regional level, based on exogenously defined data from other integrated assessment models. For our B2D climate policy cases, we use a trajectory based on an SSP2 RCP2.6 scenario from the IMAGE model[44] which leads to global net negative $CO_2$ emissions from LULUCF from 2060 onwards. In our NDC cases, we draw on an average of four SSP2 RCP6 scenarios from the IMAGE, MESSAGE, REMIND and WITCH models[45].

Future demands for energy services (including mobility, lighting, residential, commercial and industrial heat and cooling) are exogenously defined and drive the evolution of the system so that energy supply meets demands throughout the time horizon. We use energy service demands derived from SSP2[46]. For the B2D cases, we run the model with an elastic demand function, with energy service demands reducing as the marginal price of satisfying the energy service increases. Decisions around what energy sector investments to make across regions are determined based on the cost-effectiveness of investments, taking into account the existing system today, energy resource potential, technology availability, and crucially policy constraints such as emissions reduction targets. The model time horizon runs to 2100, in line with the timescale typically used for climate stabilisation. The model produces results in 5 year steps (or milestones). This level of granularity is necessary as it represents a trade-off to keep the model computationally tractable. The milestone blocks are centred on the reported years (e.g., 2025 captures 2023–2027 and 2030 captures 2028–2032) and within those blocks the results are stepwise. The model is calibrated based on 2005 IEA energy balances. Additional constraints have been introduced in the model to help represent the energy system in 2010, 2015 and 2020. The social discount rate used in the calculation of net present value (used as the basis for the objective function) is set at 3.5%.

In conjunction with a cumulative $CO_2$ budget, we place an upper limit on annual $CH_4$ and $N_2O$ emissions in our B2D scenarios based on pathways from the IPCC's Special Report on 1.5 °C scenario database[47]. We select all pathways that have warming at or below 2 °C in 2100 and take an average across these scenarios to derive a $CH_4$ and $N_2O$ emissions trajectory that is in line with a 2 °C world. Further information on key assumptions used in the model is provided in SI section 3. The TIAM-UCL model version used for this analysis was 4.1.2, and was run using TIMES code 4.5.6 with GAMS 38. The model solver used was CPLEX 20.1.0.1.

The model has been subject to diagnostic studies for comparison with other Integrated Assessment Models across a range of indicators. More details can be found in Harmsen et al.[48].

## Reporting summary

Further information on research design is available in the Nature Portfolio Reporting Summary linked to this article.

## Data availability

The results data presented in the figures are provided in the file supplementary data 1. The processed model input data can be found in the model database, which can be found on Zenodo at this link- https://zenodo.org/records/14098070. Given the complexity of the model, further guidance will be provided on model assumptions upon reasonable request from the corresponding author. Note however that key source data are also included in this paper, notably in the supplementary information. Model results files from the different scenarios are also provided at the same Zenodo link.

## Code availability

The underlying code (mathematical equations) for the model is available via GitHub (https://github.com/etsap-TIMES/TIMES_model).

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

## Acknowledgements

For M.B., C.K., J.P., P.D., and D.W., this work has been supported by the UK Energy Research Centre Phase 4 (grant number EP/S029575/1). For S.P. and J.P., their involvement was also supported by the Horizon Europe R&I programme project DIAMOND (grant no. 101081179). S.P. was also supported by the FCDO-funded Climate Compatible Growth (CCG) Programme.

## Author contributions

M.B., S.P., and J.P. designed the study, with contributions from P.D. and C.K. J.P., D.Z., and S.P. conducted the TIAM-UCL modelling with contributions from D.W. D.Z. led on presentation of modelling results. J.S. provided data on European gas flows. M.B. and S.P. led on draughting of manuscript, with contributions from all other authors. J.P. and D.Z. led in compiling the supplementary information.

## Competing interests

The authors declare no competing interests.
