## [Transparent Peer Review file · Nature Communications]

The global implications of a Russian gas pivot to Asia

Corresponding Author: Dr Steve Pye

Version 0:

Reviewer comments:

Reviewer #1

(Remarks to the Author)

The manuscript "The global implications of a Russian gas pivot to Asia" provides an assessment of the impacts of the Russian-Ukraine war on the natural gas markets and particularly the natural gas trade between Russia and China and India. In principle, the study's approach seems fitting and well performed. However, I have 3 main concerns:

- 1) The study does not provide many novel insights compared to the existing literature. The IEA's 2022 WEO (<https://www.iea.org/reports/world-energy-outlook-2022>) provided a relatively extensive analysis of the impacts of the fossil trade barriers between Russia and the EU, as did Liu et al. in the early stages of the war (<https://www.nature.com/articles/s41558-023-01606-7>). One main insight, that Russia is currently struggling with exports to India and China, can also be directly observed in real-world developments. The authors do a good job in bringing in geopolitics and interpreting the implications of a more Asia-oriented gas export. However, this is not a completely new approach. Many IAM model-based studies are based on narratives rooted in observed global developments and many studies place model results in a global political perspective.
- 2) It is a single model study. Therefore, it is difficult to assess how model assumptions influence the results.
- 3) The study is based on a large share of grey (non-peer-reviewed) literature.

Further minor comments:

I would suggest to provide key quantitative results in the abstract.

Figure 1 is based on external data, so I don't think it's necessary to show it as a key graph in the main text. In principle, the shift in exports can also be explained (as has been done) and the figure could move to the SI.

The scenarios represent REPowerEU, EU policy that is very relevant to this study. However, it is unclear if other relevant FitFor55 elements are also represented, such as the (ETS) GHG reduction targets and energy efficiency targets. It would be good to elaborate on this.

Reviewer #2

(Remarks to the Author)

Dear authors,

The results you find in the paper are surely noteworthy and will contribute to the relevant literature.

I have some minor comments, here below:

- Changed the sentence "EU Governments still spent over €651bn..." with "EU Governments still earmarked over €540bn..." (the dataset was revised last year).
- In Figure 2 you name only some of the major gas fields, why not name them all?
- In the scenario analysis you write "The first, Limited Markets (LM), sees a protracted conflict in Ukraine rule out prospects for a return of flows through pipelines from Russia to Europe, resulting in a permanent halt of all gas trade (pipeline and LNG) with Russia by 2030." I would already make clear that the EC target is doing so by 2027, without waiting to do so further down in the text.

- Your scenario P2A, sees "a lack of resolution around Russia's occupation of territory in Eastern Ukraine and Crimea" but 40bcm of EU gas imports via Ukraine. This seems highly unlikely given that the imports last year were 13.5 bcm and given that Naftogaz has signalled that it will not renew the contract to transit Russian natural gas to the EU at the end of 2024. I suggest that you either lower that figure or change the assumptions around the P2A scenario.
- There must be a typo (missing the word greater?) in the sentence "LNG tends to take a market share than pipeline imports from Central Asia particularly in LM."
- You should better explain, and possibly back with some literature (you can take a look at <https://carnegieendowment.org/politika/89552>), why in your opinion China has a stronger negotiating position to agree to a lower price than was available to Russia when exporting to Europe. Given that the volumes to be traded with China are lower than those delivered to Europe, one might argue that Europe had a stronger monopsony power than China on pipeline gas imports from Russia.
- In the sentence "Routes through Ukraine, which supplied over 80 bcm in 2019, become increasingly challenging to operate due to the conflict and ongoing contractual disputes between Russia and Ukraine, and decline from 20 bcm to zero flow by 2030" you need to specify in what year they fell to 20 bcm.
- I think you should strengthen the part on the "implication for gas markets". As it stands now is quite weak and does not provide any quantifiable indication on the price evolution in your different scenarios, neither for pipeline gas nor for LNG. You should also most prominently feature the supply forecasts for the future, as LNG capacity coming online in the next two years will be substantial.

Congratulations on this nice piece.
Giovanni Sgaravatti

Reviewer #3

(Remarks to the Author)

Overall, this is a good and a very timely paper, examining a significant issue, and applying scenario modelling to project fundamental developments related to (Russian) gas.

My main question is about emphasizing the original findings of the article. The paper could be clearer, and possibly more ambitious, in terms of key implications of its findings. Key conclusions (such as Russia may not manage to compensate for losing its European market; Europe will demand more LNG; Russia will sell more to China) are widely expected based on well-known developments. The paper could put more emphasis on the less expected possibilities. Otherwise, the paper would appear less rigorous despite being quite distinctive in terms of applying a fairly sophisticated scenario modelling.

Another major point that would be great to address is about the LM and Pivot to Asia scenarios. One could easily assume that Russia would be under greater pressure to strike new deals with China under the LM scenario due to the Ukraine war. It's not clear why sales to Asia rise under the "pivot" scenario which assumes that the Ukraine conflict is ended (even if that's not on Europe's ideal terms)

Additional points. The paper incorporates well energy market fundamentals, though one area that could benefit from more clarity is talking about key "drivers and assumptions" for global LNG. Another area that seems to be missing is "politics" (apart from the Ukraine conflict). At a minimum, the paper could spell out its key assumptions on politics (e.g. What happens to Russian politics? When is Putin replaced and how? Is there political/military escalation in US-China relations etc). A narrative on these assumptions may not be necessary, and yet the reader would benefit from clarity about the underlying thinking.

The timeframe of the scenarios: it's not initially clear. Statements such as "a permanent halt of all gas trade ... with Russia by 2030" remain vague in terms of how this plays out year after year. (lines 105-112)

Lines 36-37: I'd be cautious on predicting lower gas prices after 2026. Overall, any forecasts about gas prices would need to spell out the underlying assumptions and the baseline to be compared to. Lines 260-1: The paper suggests that Russian gas revenues will drop in all scenarios. This needs to clarify the baseline, and tell the reader key assumptions about gas prices. Why, for instance, a major gas price spike is not likely in the next 20+ years?

Lines 173-4: the estimates 55 bcm, assuming Russian gas exports to Europe return to pre-war level could be double checked in terms of Ukrainian pipeline capacity used in 2010s (in addition to the gas transit agreement with Ukraine through 2024)

Lines 203-11: The estimates gas demand in China in 2020 and 2050 appears to rounded very much (300 bcm, 500 bcm). A rigorous model is likely to find out that China will probably end up not with a round number. Such round figures, unless reached through a very methodical approach, make the model less credible.

Lines 227-229: The counterfactual scenario assumes that Russian gas exports will keep declining. This is very different from pre-war scenarios, where Russia was assumed to keep exporting larger volumes, even if European volumes were kept

constant or started to fall. This calls for an explanation about why the counterfactual scenario is so “negative” on Russian exports.

Lines 250-2: The paper suggests that the future of LNG supply will depend on large producers such as US and Qatar. This appears to ignore the emerging reality of growing number of countries interested in LNG exports.

Version 1:

Reviewer comments:

Reviewer #1

(Remarks to the Author)

The authors have done a really good job in responding to the reviewers' comments and I would recommend publishing the study in NatComms. I have two minor last comments that could hopefully be easily addressed in the final version of the paper:

1) On comment 1.3, on it being a single-model study. The authors have performed a sensitivity analysis, which is a great solution. I have one additional suggestion. It would be good to briefly reflect on the behavior of TIAM-UCL compared to other IAMs. This does not have to be an additional analysis, but can for instance be based on diagnostic studies, focusing on indicators most relevant to this study.

2) Ukraine is written incorrectly in Figure 2

Reviewer #2

(Remarks to the Author)

The authors addressed most of my comments and provided sound justification in the two instances when they decided not to take onboard my remarks. I am satisfied with the current draft.

Reviewer #3

(Remarks to the Author)

The authors have made significant changes, responding successfully to feedback.

There is only one minor suggestion for the authors: The paper argues that China has a more diversified supply base than Europe, which provides it more of a bargaining power. The authors have responded to a query on this, but I believe one could be more cautious factually. The metrics for a diversity of supply are not clear enough. For each example provided for non-Russian gas for China in the paper, one can find even more examples regarding Europe's diverse supplies (Norway, Africa, Caspian). Likewise for LNG. Europe taken as a whole still has more intake capacity than China.

Overall, good job in improving the paper. I would be happy to see this paper published.

NCOMMS-24-17473-T The global implications of a Russian gas pivot to Asia

Responses to reviewer comments

Reviewer 1

Comment No.	Reviewer's comment	Authors' response
1.1	The manuscript "The global implications of a Russian gas pivot to Asia" provides an assessment of the impacts of the Russian-Ukraine war on the natural gas markets and particularly the natural gas trade between Russia and China and India. In principle, the study's approach seems fitting and well performed.	We thank the reviewer for this positive comment, and for the constructive feedback provided.
1.2	However, I have 3 main concerns: 1) The study does not provide many novel insights compared to the existing literature. The IEA's 2022 WEO (https://www.iea.org/reports/world-energy-outlook-2022) provided a relatively extensive analysis of the impacts of the fossil trade barriers between Russia and the EU, as did Liu et al. in the early stages of the war (https://www.nature.com/articles/s41558-023-01606-7). One main insight, that Russia is currently struggling with exports to India and China, can also be directly observed in real-world developments. The authors do a good job in bringing in geopolitics and interpreting the implications of a more Asia-oriented gas export. However, this is not a completely new approach. Many IAM model-based studies are based on narratives rooted in observed global developments and many studies place model results in a global political perspective.	Thank you for raising these points / concerns. We recognise that the two studies mentioned do consider the implications of the EU gas crisis. However, we believe that our study is somewhat distinctive, in terms of the approach taken and in relation to the scenarios developed. On approach, and the incorporation of geopolitics, a key motivation comes from a review paper by Blondeel et al. 2023 which reviewed other global scenarios, and highlighted that the implementation of geopolitics into scenarios is limited. While geopolitics may often be considered in model narratives, the model implementation is often missing or very partial. Similarly, the community working on the geopolitics of transitions often omit any quantitative analysis e.g. Giuli and Oberthur (2023) & Scholten et al (2020) to name two examples. We believe that this paper, bringing together political science / economic geography disciplines with energy and climate modellers, marks an advancement in demonstrating how geopolitics can be better reflected in global scenarios. This advancement has described in the paper based on the addition of the following edited text - 'This research is a collaboration between social scientists interested in energy geopolitics and global gas security and energy systems modellers focused on quantifying the role of fossil fuels in the energy transition. Our approach deploys a geopolitical framing to inform the development of scenarios about Russia's future role in global gas markets, the implications of which are assessed using the TIAM-UCL Integrated Assessment Model (IAM) (see Methods). The novel aspect is the

		dynamic process of creating scenarios informed by an interdisciplinary approach. This allows for an exploration of the interrelationships between various Russian gas supply scenarios and different climate change futures to assess the impact of a Russian gas pivot to Asia for global energy security and climate action.' On the scenarios constructed, our key focus has been to develop alternative scenarios on how the Russian export strategy might evolve, depending on a range of factors. In contrast, the IEA's approach is to consider the emerging energy context following the crisis as it applies to its standard set of scenarios. Our approach provides a different perspective, with what we believe to be new insights, focused on the interrelated demand-supply relationships between Russia, Europe and China. These have been further developed in this revision, based on the previous scenarios plus new sensitivities, and include -  ● Potential for Russia to partially regain markets for exports - but subject to high uncertainty e.g. impact of climate policy, Chinese energy security strategy, LNG export capacity build out etc. ● Europe being more exposed to increased LNG dependency (and market exposure) - but mitigating that to some extent by declining demand in the longer term ● China having some challenging choices to make on energy security e.g. the balance between domestic production, LNG or pipelines - all of which have implications for Russia and wider gas market. The Liu paper is interesting and an important contribution but focuses heavily on the macroeconomic impacts based on different levels of disruption of Russian exports. There has limited consideration of the implications of a shift towards Asian markets, with China not explicitly mentioned in the analysis set-up. Conversely, our analysis looks in greater detail at the different configurations of pipeline capacity into China - and uncertainty around demand in both European, Asian and other markets.
1.3	2) It is a single model study. Therefore, it is difficult to assess how model assumptions influence the results.	Thank you for the comment. We note that there are many impactful single modelling studies across Nature and other journals (e.g. Muttitt et al. 2023 ; Welsby et al. 2021). However, we recognise that this paper would be strengthened by additional sensitivity analysis. We have taken on board this concern by expanding our analysis

		to focus on some key assumptions embedded in our scenarios - to explore the impact that they have on the modelling results. We have developed the sensitivity analysis around our central scenarios, LM and P2A, to test key assumptions underpinning the scenarios, to better understand implications of the results, notably on Russian exports (both positive and negative) and the wider market. This provides a more rigorous assessment of potential uncertainties embedded in our scenario assumptions. The sensitivity cases are run across all scenarios, and include the following – Reduced Chinese domestic production (supply case): An important assumption in our core scenarios is that China will maintain the share of domestically produced gas as a percentage of total gas consumption. This is set at 50% based on current Chinese policy. Such an assumption influences the required level of imports so is important to explore. In this sensitivity case, we have relaxed this to a maximum domestic production level of 30%, which is aligned with the policy on the required domestic oil production. Increased Chinese gas demand (demand case): The demand for gas in our scenarios is, by design, largely driven by the climate policy ambition assumed. However, this does not necessarily capture the full range of possible gas demand levels in China, a key market which has implications for regional and global production. For this sensitivity, we have reviewed the range of gas demand levels in China (based on a review of the scenario literature, SI section 1) and implemented a higher level of demand in China. We developed a case whereby we increased the amount of natural gas demand into the economy, primarily by relaxing constraints on uptake in different sectors. This resulted in 2030 supply levels in China reaching 515-525 Bcm in B2D cases compared to the previous level of 455 Bcm, and 415-460 BCM in NDC cases compared to the previous level of 345 Bcm. Constraints on Russian LNG exports (supply case): A key uncertainty concerns the ability of Russia to ramp up its LNG export business in the medium to long term, notably in LM where export routes via pipelines are somewhat constrained. Currently in LM we have assumed some level of near term constraint based on sanctions but with fewer restrictions in the
--	--	---

		medium to longer term. In the near term this includes LNG exports from Russia to the G7 and Europe being prohibited from 2030 in LM. This sensitivity brings more stringent restrictions on the ability of Russia to build out its LNG exports in the medium to long term. Our new medium to long term constraint on the ramp up of Russian LNG sees at most a 2% per year increase in export flows, placing much more friction on its growth. It is largely based on an outcome whereby insufficient ice-class carriers are supplied to enable full export from Yamal LNG and Arctic LNG due to western sanctions, compounded by EU sanction on the transshipment of cargoes destined for non-EU markets. This new analysis is described and presented in SI section 3, with cross-references to the implications in the main paper.
1.4	3) The study is based on a large share of grey (non-peer-reviewed) literature.	We acknowledge the high proportion of literature from a range of research organisations and think tanks. We feel that given the topic in question, reference to this contemporary literature is absolutely necessary. In addition, this literature is primarily used to set the geopolitical context, and not to inform methodology and research gaps. We therefore do not consider this to be a weakness but rather a necessity given the objective of the research.
1.5	Further minor comments: I would suggest to provide key quantitative results in the abstract.	We have added the following line in the abstract - 'Compared to 2020, Russia's gas exports are down by 31-47% in 2040 where new markets are limited and 13-38% under a pivot to Asia strategy.' A limit of 150 words for the abstract means that we had to also delete the following sentence - 'Alongside Europe's approach to meet future demand such decisions are likely to impact LNG markets and future price volatility.'
1.6	Figure 1 is based on external data, so I don't think it's necessary to show it as a key graph in the main text. In principle, the shift in exports can also be explained (as has been done) and the figure could move to the SI.	Whilst we agree that the text can convey this, our preference would be to keep this in the main body of the paper just because it provides a helpful visual of the price volatility in recent years and the rapid decline in pipeline exports.
1.7	The scenarios represent REPowerEU, EU policy that is very relevant to this study. However, it is unclear if other relevant FitFor55 elements are also represented,	Our NDC climate policy case captures the EU NDC as of November 2021 which included a 55% reduction in GHG emissions from 1990 levels by 2030. With regard to energy efficiency targets,

	such as the (ETS) GHG reduction targets and energy efficiency targets. It would be good to elaborate on this.	our representation of REPowerEU requires a minimum of a 13% reduction in final energy consumption in the buildings and transport sectors. In Methods, in the section relevant to REPowerEU, we have added - 'We also include the EU NDC as of November 2021 which included a 55% reduction in GHG emissions from 1990 levels by 2030, and the energy efficiency targets associated with REPowerEU, which require a minimum of a 13% reduction in final energy consumption in the buildings and transport sectors.'
--	--	--

Reviewer 2

Comment no.	Reviewer's comment	Authors' response
2.1	The results you find in the paper are surely noteworthy and will contribute to the relevant literature.	We thank the reviewer for this positive comment, and for the feedback provided.
2.2	I have some minor comments, here below: - Changed the sentence "EU Governments still spent over €651bn..." with "EU Governments still earmarked over €540bn..." (the dataset was revised last year).	The text has been edited to reflect this different figure. We understand that €651bn included other non-EU European countries.
2.3	In Figure 2 you name only some of the major gas fields, why not name them all?	Thank you for the suggestion. The figure has now been updated with all major gas field names (see Figure 1 in the revised manuscript).
2.4	In the scenario analysis you write "The first, Limited Markets (LM), sees a protracted conflict in Ukraine rule out prospects for a return of flows through pipelines from Russia to Europe, resulting in a permanent halt of all gas trade (pipeline and LNG) with Russia by 2030." I would already make clear that the EC target is doing so by 2027, without waiting to do so further down in the text.	In the text, '2030' has been replaced by '2027 (as per the EU target)'.
2.5	Your scenario P2A, sees "a lack of resolution around Russia's occupation of territory in Eastern Ukraine and Crimea" but 40bcm of EU gas imports via Ukraine. This seems highly unlikely given that the imports last year were 13.5 bcm and given that Naftogaz has signalled that it will not renew the contract to transit	Given the comment by the reviewer and based on further reflection, we have aligned European pipeline gas imports via Ukraine with our LM scenario. This seems reasonable given what appears to be limited prospects for renewal of a transit agreement. The relevant text in the manuscript has therefore been changed to -

	Russian natural gas to the EU at the end of 2024. I suggest that you either lower that figure or change the assumptions around the P2A scenario.	'Given this context, Europe remains unwilling to revert to its previous dependence on Russian energy, with no pipeline gas via Ukraine after 2025 due to the non-renewal of the transit agreement, and a maximum 15 bcm via TurkStream.' The P2A narrative in the Methods section has also been updated to reflect this change. This is also now reflected in the modelling of this scenario too (see changes to results in SI section 2). On TurkStream, we have also considered that this supply does not expand. This is based on the bottleneck on the Serbia-Hungary border, where capacity is 23 MMcm/d (8.4 bcma). So, if / when transit via Ukraine stops, there is very limited spare capacity on that border to bring additional volumes of Russian pipeline gas to Central Europe via Turkish Stream. Rather, Gazprom is facing a challenge in holding onto its market share in SE Europe, rather than increasing its sales in that region. For that reason, we have assumed no increase in Gazprom exports via Turkish Stream even if Ukrainian gas transit does cease. This additional thinking has been further reflected in the narrative in the Methods part of the paper.
	There must be a typo (missing the word greater?) in the sentence "LNG tends to take a market share than pipeline imports from Central Asia particularly in LM."	An edit has been made, with 'higher' added before the word 'market'.
2.6	You should better explain, and possibly back with some literature (you can take a look at https://carnegieendowment.org/politika/89552), why in your opinion China has a stronger negotiating position to agree to a lower price than was available to Russia when exporting to Europe. Given that the volumes to be traded with China are lower than those delivered to Europe, one might argue that Europe had a stronger monopsony power than China on pipeline gas imports from Russia.	China imports pipeline gas from Turkmenistan, Russia, Myanmar, Kazakhstan, and Uzbekistan (although the latter is struggling to maintain its exports). China also imports LNG from a wide variety of countries (mostly Australia, Qatar, Russia, Malaysia, Indonesia, the United States, and Papua New Guinea, who accounted for 90% of China's LNG imports in 2023). Indeed, China was the world's largest LNG importer in 2021 before its imports fell back in 2022 and only partially recovered in 2023. Therefore, China has a diverse supply portfolio, and in the context of a substantial growth in global LNG supply forecast for 2025-2030, is in no rush to sign binding new pipeline supply agreements with Russia. By contrast, the decline in Russian pipeline gas exports to Europe following Russia's invasion of Ukraine has left a substantial volume of Russian gas production effectively 'trapped' in Russia, with no alternative pipeline export markets (save

		for the construction of Power of Siberia 2 for export to China). Western sanctions are also preventing the ramp-up of Russia's LNG exports, as seen at Arctic LNG 2. (See WITS database for gas supplies to China). In the final section of the manuscript we have added the following text 'This stronger position is based on China's more diversified pipeline supply and large LNG import potential (based on physical capacity and contracting).' This follows this sentence to which we understand the reviewer to be commenting on - 'First, China has a strong negotiating position to agree a lower price than was available to Russia when exporting to Europe, meaning that even if gas exports were increased, revenues would not recover.' We are aware of Sergei Vakulenko's work at Carnegie and have referenced it. We also reference the work of Tatiana Mitrova and colleagues at Columbia, all of which supports the view that Russia is in a relatively weak position when it comes to negotiating an agreement on PoS 2. It is also telling that no agreement has, so far, been reached.
2.7	In the sentence "Routes through Ukraine, which supplied over 80 bcm in 2019, become increasingly challenging to operate due to the conflict and ongoing contractual disputes between Russia and Ukraine, and decline from 20 bcm to zero flow by 2030" you need to specify in what year they fell to 20 bcm.	Transit fell to 20.4 bcm in 2022, as measured on the Russia-Ukraine border. Data from ENTSOG Transparency Platform. https://transparency.entsog.eu/#/map The text has been edited accordingly.
2.8	I think you should strengthen the part on the "implication for gas markets". As it stands now is quite weak and does not provide any quantifiable indication on the price evolution in your different scenarios, neither for pipeline gas nor for LNG. You should also most prominently feature the supply forecasts for the future, as LNG capacity coming online in the next two years will be substantial.	We prefer not to provide any specific price trajectories for this modelling, due to some limitations with price formation in the model. The model, whilst producing shadow prices based on a cost only basis, does not fully represent the regionalised markets for gas, the different contracting structures for LNG, or fiscal regimes. We view the shadow price produced in the model as a simplified proxy for decisions based on cost - but not one to publish as a market price. We are also focused on the longer term, whereby suggesting an outlook for market prices is extremely challenging. On the latter point, the projections of LNG are provided in SI Figure 3, under different scenarios (so not forecasts). We should also state that these

		do reflect existing capacity, and that which is planned over the next 5 years. Recent LNG capacity expansion, and the new liquefaction and regasification capacity (including capacity under construction until 2030) has been incorporated into this work, based on data from the IGU LNG Report 2024.
--	--	---

Reviewer 3

Comment no.	Reviewer's comment	Authors' response
3.1	Overall, this is a good and a very timely paper, examining a significant issue, and applying scenario modelling to project fundamental developments related to (Russian) gas.	We thank the reviewer for this positive comment, and for the excellent feedback provided.
3.2	My main question is about emphasizing the original findings of the article. The paper could be clearer, and possibly more ambitious, in terms of key implications of its findings. Key conclusions (such as Russia may not manage to compensate for losing its European market; Europe will demand more LNG; Russia will sell more to China) are widely expected based on well-known developments. The paper could put more emphasis on the less expected possibilities. Otherwise, the paper would appear less rigorous despite being quite distinctive in terms of applying a fairly sophisticated scenario modelling.	We agree with the author that we needed to bring out more clearly the implications of the modelling analysis in our manuscript. However, we also think that while some of the findings may appear intuitive (as indicated by the reviewer) the purpose of this approach (as reflected in the paper) is to join geopolitical analysis of transitions and energy modelling; for the community engaged in geopolitics, this helps to bring coherency and consistency into narratives and better determine the magnitude of impacts and wider system effects, notably given the interconnectedness of gas supply and demand. And for modelling, this is to better capture geopolitical narratives, by better parameterising models to reflect real world realities, and move away from more normative type analyses. To address this valid comment more broadly, including developing the insights and increasing analytical rigour, we have provided a more rigorous assessment of some of the key assumptions via sensitivity analysis (as outlined in our response to reviewer 1, comment 1.3). We have also developed the results section to bring out some more of the key findings / insights.
3.3	Another major point that would be great to address is about the LM and Pivot to Asia scenarios. One could easily assume that Russia would be under greater pressure to strike new deals with China under the LM scenario due to the Ukraine war. It's not clear why sales to Asia rise under the "pivot" scenario which assumes that the Ukraine conflict is ended (even if that's not on Europe's ideal terms)	It is true that under LM, stronger restrictions on gas trade with Europe could put pressure on Russia to strike a deal with China. However, in LM we assume China does not want to expand pipeline capacity because of energy security considerations. Given its observations of how the Russia-European supply crisis unfolded, there is a hesitancy to increase supply via a single supplier, with a stronger push towards domestic production and LNG.

		We have made this more explicit in our scenario narrative, both in the short version in the main paper and in the Methods. In the short version (first paragraph under subheading Future scenarios of Russian gas exports), we have added the following text - 'This reflects China's position on energy security; avoiding increased dependency given the Russia-European supply crisis, and a focus on further diversification based on domestic production and LNG.' In the longer version of the narrative in the Methods section, we have made the following edit - 'Under this scenario, China stalls on agreeing an investment decision on the Power of Siberia 2 pipeline, limiting Russia's options further. This is based on Chinese energy security concerns related to increased dependence on Russian pipeline gas, given the recent European supply crisis, and a strategy towards increased diversification of supply via domestic production and multiple LNG contracts.'
3.4	Additional points. The paper incorporates well energy market fundamentals, though one area that could benefit from more clarity is talking about key “drivers and assumptions” for global LNG. Another area that seems to be missing is “politics” (apart from the Ukraine conflict). At a minimum, the paper could spell out its key assumptions on politics (e.g. What happens to Russian politics? When is Putin replaced and how? Is there political/military escalation in US-China relations etc). A narrative on these assumptions may not be necessary, and yet the reader would benefit from clarity about the underlying thinking.	On the point regarding key drivers and assumptions for global LNG, the modelling endogenises the contribution of this supply, of course accounting for current and planned capacity (liquefaction & regasification) in different regions, and current flows. How quickly this can build out is subject to constraints. The overall contribution will be a function of overall gas demand - and the various constraints on pipeline supply and existing / build out of LNG capacity. Political uncertainties, including but not limited to China-Russia and China-US relations, have a significant impact on the LNG trade landscape. However, the LNG market, being more flexible compared to other gas trade markets, is highly sensitive to demand. When examining the key driver—Chinese LNG demand—it is important to note that, prior to 2020, China relied on a mix of spot and short-term LNG contracts due to their flexibility and lower cost. However, by the end of 2020, LNG spot prices surged, primarily due to severe weather conditions in Northeast Asia and the subsequent crisis following Russia's invasion of Ukraine, which exacerbated market tightness. Chinese buyers found themselves exposed to high spot market price risks and uncertainties in spot

		LNG supply. Consequently, they began seeking more long-term contracts from LNG suppliers. In 2021, China’s LNG contracting activity surged, with 20 new contracts representing approximately 40 percent by volume of all new LNG contracts signed (GIIGNL, 2022). Most of these contracts have a duration of 10-20 years, and among the contracts signed in 2021, about half by volume are for US LNG, with 20% from Qatar (Corbeau and Yan, 2022). This array of LNG contracts provides China with a more diversified and relatively stable LNG supply portfolio during a time of strained relations with major LNG exporters, such as the US and Australia. As a result, the impact of political uncertainties on the LNG trade landscape is expected to be significantly reduced over the next decade. Therefore, this paper does not make further political assumptions regarding LNG beyond representing the existing LNG capacity and current LNG contracts. On politics, the reviewer is right that implicitly underpinning the narratives would also be an evolving political landscape. However, this is challenging to unpick and feels to us that this would be an expansion of the scope of the paper. What we have done is be clearer on what we mean by a ‘geopolitical’ framing, which we think helps the reader better understand the scope of the paper. In the Methods part of the paper, at the start of the section on ‘scenario definition’, we have elaborated on this as follows - ‘Scenarios have been developed for this research based on interaction between economic geography and political science disciplines, with energy systems modellers. This is motivated by a recognition that such communities need to collaborate to better reflect energy geopolitics in scenarios. By energy geopolitics, we mean ‘the interaction of geographical factors, such as the distribution of centres of supply and demand, with state and non-state actors’ attempts to ensure an affordable, reliable and sustainable supply of energy’[Blondeel et al 2024].’
3.5	The timeframe of the scenarios: it’s not initially clear. Statements such as “a permanent halt of all gas trade ... with Russia by 2030” remain vague in terms of how this plays out year after year. (lines 105-112)	Thank you for the comment as we realised we could have been clearer in communicating the modelling timeframe. In summary, the model produces results in 5 year steps (milestones). This level of granularity is necessary as it represents a trade-off to keep the

		model computationally tractable. Therefore, in practice a rapid decline in Russia to Europe gas trade is constrained into the model in 2025 (from the level recorded in 2020) which then reaches 0 in 2030. We have added the following text in the Methods section (in subsection The TIAM-UCL model - 'The model produces results in 5 year steps (or milestones). This level of granularity is necessary as it represents a trade-off to keep the model computationally tractable. The milestone blocks are centred on the reported years (e.g. 2025 captures 2023-2027 and 2030 captures 2028-2032) and within those blocks the results are stepwise.'
3.6	Lines 36-37: I'd be cautions on predicting lower gas prices after 2026. Overall, any forecasts about gas prices would need to spell out the underlying assumptions and the baseline to be compared to.	We accept the point and have softened the language used. The sentence now reads - 'Continued price volatility is likely until 2026, when a surge in new LNG supply from the US and Qatar could lower global prices, potentially enabling Europe to secure LNG imports at lower prices as it shuns Russian gas, including LNG.' We have replaced the word 'should' with 'could', and added 'potentially enabling' to replace 'and enable'. To note additionally, we see an industry consensus emerging that it will be an oversupplied buyers' market by 2026 through to 2030. Our assumptions would be: 1. massive surge in LNG supply, 2. Falling European demand, 3. Uncertainty over demand growth in Asia. The net result is an oversupplied market and lower prices than has recently been the case. High prices and volatility are likely to be with us until 2026 at least.
3.7	Lines 260-1: The paper suggests that Russian gas revenues will drop in all scenarios. This needs to clarify the baseline, and tell the reader key assumptions about gas prices. Why, for instance, a major gas price spike is not likely in the next 20+ years?	This comment refers to the sentence 'In all scenarios, Russia has reduced exports and the Russian Government has reduced revenues.' To be more precise, the sentence has been changed to - 'In all scenarios, Russia has reduced exports relative to today and a 'no crisis' counterfactual, resulting in lower revenues for the Russian Government.' It is true that further crises might emerge in the future which these scenarios do not capture. We have therefore based this comment on the scenario narratives which have been developed.

3.8	Lines 173-4: the estimates 55 bcm, assuming Russian gas exports to Europe return to pre-war level could be double checked in terms of Ukrainian pipeline capacity used in 2010s (in addition to the gas transit agreement with Ukraine through 2024)	Refer to our response to comment 2.5, where we explain that we have revised the assumptions in our narrative.
	Lines 203-11: The estimates gas demand in China in 2020 and 2050 appears to rounded very much (300 bcm, 500 bcm). A rigorous model is likely to find out that China will probably end up not with a round number. Such round figures, unless reached through a very methodical approach, make the model less credible.	In response to this comment, we have added in the more detailed estimates based on the modelling. Our previous approach was to use more rounded values, recognising the uncertainty and wanting to avoid false precision. This was also done by using words such as ‘around’ or ‘over’ prior to the values in the text, to indicate they were not the specific values from the modelling.
	Lines 227-229: The counterfactual scenario assumes that Russian gas exports will keep declining. This is very different from pre-war scenarios, where Russia was assumed to keep exporting larger volumes, even if European volumes were kept constant or started to fall. This calls for an explanation about why the counterfactual scenario is so “negative” on Russian exports.	In Fig 4a we see that NDC_REF has Russian exports rising to 2040 and only then beginning to decline. And even then, 2050 sees roughly the same export volumes as 2020, albeit with a substantially increased role for LNG and a large drop in exports to Europe via pipeline. This pathway is largely set by the combined interplay of: a) our assumption in the counterfactual that pipeline trade between Russia and China is capped at Power of Siberia 1 + Sakhalin (48 bcm total), b) our assumption that new pipeline capital costs between AFR and Europe are cheaper than Russia and Europe so around 2045/50 reinvestment (following decommissioning of some of the early capacity after our 40 year economic lifespan assumption) happens between the former trading pair and c) European decarbonisation under its NDC commitments see a ~30% cut in gas demand by 2050 relative to 2020.
	Lines 250-2: The paper suggests that the future of LNG supply will depend on large producers such as US and Qatar. This appears to ignore the emerging reality of growing number of countries interested in LNG exports.	The text in the paper states that ‘If China were to take a more diversified supply option, this will have implications for the LNG market, with potential for market tightening, although of course this will also be dependent on large suppliers such as USA and Qatar.’ USA and Qatar are therefore provided as examples of large (and important) LNG producers, as opposed to the only market players that matter. It is the case that US and Qatari expansion account for the bulk of the coming capacity increase, with Qatar supplying based on the traditional model based on long-term

		contracts and destination clauses, while the US supplies flexibility in terms of pricing and destination.
--	--	---

Response to reviewers' comments - NCOMMS-24-17473A

Reviewer	Comment	Authors' response
1	The authors have done a really good job in responding to the reviewers' comments and I would recommend publishing the study in NatComms. I have two minor last comments that could hopefully be easily addressed in the final version of the paper: 1) On comment 1.3, on it being a single-model study. The authors have performed a sensitivity analysis, which is a great solution. I have one additional suggestion. It would be good to briefly reflect on the behavior of TIAM-UCL compared to other IAMs. This does not have to be an additional analysis, but can for instance be based on diagnostic studies, focusing on indicators most relevant to this study. 2) Ukraine is written incorrectly in Figure 2	In response to comment 1, TIAM-UCL was included in a diagnostic study, by Harmsen et al. 2021 – and compared across a range of indicators. However, none of the indicators used in this study are easily compared with the set used for diagnostic. Therefore, the approach we have taken is to mention the diagnostic study at the end of the methods study, and refer the reader there for more information regarding model behaviour. In response to comment 2, this has been corrected.
2	The authors addressed most of my comments and provided sound justification in the two instances when they decided not to take onboard my remarks. I am satisfied with the current draft.	No changes have been made here
3	The authors have made significant changes, responding successfully to feedback. There is only one minor suggestion for the authors: The paper argues that China has a more diversified supply base than Europe, which provides it more of a bargaining power. The authors have responded to a query on this, but I believe one could be more cautious factually. The metrics for a diversity of supply are not clear enough. For each example provided for non-Russian gas for China in the paper, one can find even more examples regarding Europe's diverse supplies (Norway, Africa, Caspian). Likewise for LNG. Europe taken as a whole still has more intake capacity than China. Overall, good job in improving the paper. I would be happy to see this paper published.	In response we have removed the word 'more' from the following sentences - This stronger position is based on China's more diversified pipeline supply and large LNG import potential (based on physical capacity and contracting). China's position of strength in negotiations comes from its more diversified pipeline supply, large LNG import potential and stable domestic production outlook.